# Synthesis of TiO_2_/WO_3_ Composite Nanofibers by a Water-Based Electrospinning Process and Their Application in Photocatalysis

**DOI:** 10.3390/nano10050882

**Published:** 2020-05-02

**Authors:** Vincent Otieno Odhiambo, Aizat Ongarbayeva, Orsolya Kéri, László Simon, Imre Miklós Szilágyi

**Affiliations:** 1Department of Inorganic and Analytical Chemistry, Budapest University of Technology and Economics, Szent Gellért tér 4., H-1111 Budapest, Hungary; ayzatonga@gmail.com (A.O.); orsolyakeri@gmail.com (O.K.); imre.szilagyi@mail.bme.hu (I.M.S.); 2Department of Organic Chemistry and Technology, Budapest University of Technology and Economics, Budafoki út 8., H-1111 Budapest, Hungary; simon.laszlo92@gmail.com

**Keywords:** electrospinning, polyvinylpyrrolidone, titanium(IV) bis(ammonium lactato) dihydroxide, ammonium metatungstate, nanofibers, photocatalysis

## Abstract

TiO_2_/WO_3_ nanofibers were prepared in a one-step process by electrospinning. Titanium(IV) bis(ammonium lactato)dihydroxide (TiBALDH) and ammonium metatungstate (AMT) were used as water-soluble Ti and W precursors, respectively. Polyvinylpyrrolidone (PVP) and varying ratios of TiBALDH and AMT were dissolved in a mixture of H_2_O, EtOH and CH_3_COOH. The as-spun fibers were then heated in air at 1 °C min^−1^ until 600 °C to form TiO_2_/WO_3_ composite nanofibers. Fiber characterization was done using TG/DTA, SEM–EDX, FTIR, XRD, and Raman. The annealed composite nanofibers had a diameter range of 130–1940 nm, and the results showed a growth in the fiber diameter with an increasing amount of WO_3_. The photocatalytic property of the fibers was also checked for methyl orange bleaching in visible and UV light. In visible light, the photocatalytic activity increased with an increase in the ratio of AMT, while 50% TiBALDH composite fibers showed the highest activity among the as-prepared fibers in UV light.

## 1. Introduction

The main challenges in enhancing the efficiency of the photocatalytic activity of TiO_2_ are to increase its light-absorbing range to a higher wavelength and to decrease the speed of the charge carriers to recombine [1,2,3,4,5]. Different methods of overcoming these problems have been reported. One such strategy is to prepare composites of TiO_2_ with semiconductor metal oxides that have a narrow bandgap. Tungsten oxide has been widely reported to have excellent photochromic and electrochromic properties. WO_3_ has a higher discharge capacity, a smaller bandgap, and is, therefore, a suitable compound to combine with TiO_2_ [6,7,8]_._ Many methods of coupling TiO_2_ with WO_3_ have been reported. Smith et al. prepared TiO_2_–WO_3_ composite nanotubes by anodic oxidation of their precursors [9]. Szilagyi et al. used electrospinning to synthesize WO_3_ fibers and coated them with TiO_2_ using atomic layer deposition (ALD) [10]. Reyes et al. incorporated WO_3_ into TiO_2_ nanotubes by electrodeposition [11]. Chakornpradit et al. used sol-gel and electrospinning methods to prepare TiO_2_/WO_3_ composite micro-nanofibers [12]. Epifani et al. prepared TiO_2_/WO_3_ nanocomposites by colloidal processing in solvothermal conditions [13]. The challenge of synthesizing composite TiO_2_/WO_3_ nanofibers using a single technique has been that while most TiO_2_ precursors hydrolyze in water, WO_3_ precursors are water-soluble. 

In our previous study, we used a water-soluble precursor to prepare TiO_2_ nanofibers by electrospinning [14]. The study presents the possibility of fabricating composite nanofibers from TiO_2_ and other metal oxides in a one-step synthesis using electrospinning. Electrospinning has been reported extensively as a suitable method for preparing nanofibers that are uniform and have a large surface area [15,16,17,18,19]. This technique uses an electrically charged jet of polymer solution, or melt is used to form nanofibers that are collected on a grounded target [20,21,22,23,24].

In this study, an aqueous solution of polyvinylpyrrolidone (PVP), titanium(IV) bis(ammonium lactate)dihydroxide (TiBALDH), and ammonium metatangstate (AMT) was electrospun, and the as-synthesized fibers annealed to form TiO_2_/WO_3_ composite nanofibers. We investigated how the properties of the electrospun fibers changed when different volume ratios of precursor solutions were used. A simultaneous thermogravimetry/differential thermal analysis (TG/DTA) of the electrospun fibers was done in air and nitrogen to determine the annealing temperatures. By annealing the electrospun fibers at 600 °C in air, the polymer was removed, and the precursors decomposed to form TiO_2_/WO_3_ nanofibers. The fibers were studied using Fourier transform infrared spectroscopy (FTIR), scanning electron microscopy (SEM) and energy dispersive X-ray (EDX). The specific surface area of the annealed fibers was studied by the Bruneauer–Emmett–Teller (BET) method. The annealed nanofibers were investigated by X-ray diffraction (XRD), Raman spectroscopy and UV-Vis diffuse reflectance spectroscopy as well. The bandgap of the annealed fibers was also determined. The photocatalytic activity of the fibers in UV and visible light was investigated using methyl orange.

## 2. Materials and Methods 

### 2.1. Materials

All the chemicals were of analytical grade and used as received. Polyvinylpyrrolidone (PVP, (C_6_H_9_NO)_n_, × K-90), ethyl alcohol, acetic acid, titanium(IV) bis(ammonium lactato)dihydroxide solution (50 wt % in water), and Ammonium metatungstate were obtained from Sigma Aldrich, (Budapest, Hungary).

### 2.2. Preparation and Characterization of TiO_2_/WO_3_ Fibers

The polymer solution had 20 wt % PVP dissolved in a 1:1 mixture of acetic acid and ethyl alcohol. An aqueous solution of AMT contained 1 g of AMT in 1 mL of distilled water. The solution for electrospinning was prepared by adding 2 mL of the polymer solution to 2 mL of an aqueous solution containing different TiBALDH: AMT volume ratios 100:0, 90:10, 50:50, 10:90 and 0:100. The solution was magnetically stirred at 25 °C for 6 h. For the electrospinning procedure, the solution was transferred into a 10 mL plastic syringe fitted with a needle. The voltage used was 20 kV, the flow rate was 1 mLh^−1^, and the distance between the needle and the collecting plate was 15 cm.

The thermal properties of the as-spun fibers were studied in air and nitrogen using an A STD 2960 Simultaneous DTA/TGA (TA Instruments Inc., New Castle, DE, USA) thermal analyzer. A heating rate of 10 °C min^−1^ was used, and the fibers were heated to 600 °C. The as-spun fibers were calcined in air at 600 °C for 6 h to form nanofibers. The heating rate used was 1 °C min^−1^ up to 600 °C. 

The morphology of the as-spun and annealed fibers was characterized by SEM using a JEOL JSM-5500LV (Tokyo, Japan) scanning electron microscope coupled with energy dispersive X-ray (EDX). All the fibers were covered with a thin Au/Pd layer, using a sputter coater. 

The surface area analysis of the annealed fibers was done by a multipoint Brunauer–Emmett–Teller (BET) method, using nitrogen adsorption/desorption isotherm measurements at 25 °C.

A Nicolet 6700 (Thermo fisher scientific USA) Fourier transform infrared spectrophotometer was used to obtain the FTIR spectra of the fibers in the range of 400–4000 cm^−1^ in transmittance mode.

The XRD patterns of the annealed fibers were obtained with a X’pert Pro MPD X-ray diffractometer (PANalytical, Almelo, Netherlands), using Cu K_α_ irradiation. Raman spectroscopy was done using a Jobin Yvon Labram Raman instrument (Horiba, Kyoto, Japan) with an Olympus BX41 microscope (Olympus, Tokyo, Japan) fitted with a green Nd-YAG laser. The measuring range was 72–1560 cm^−1^.

The diffuse reflectance UV-Vis absorption spectra of the annealed fibers were measured by (Avantes BV, Apeldoorn, Netherlands) with fiber optic spectrophotometer between 250 and 800 nm. The bandgap was calculated using the Kubelka–Munk theory from the Tauc plots [25].

### 2.3. Photocatalysis

The photocatalytic activity of the fibers was investigated in both UV and visible light. In a quartz cuvette, 1.0 mg of the annealed fibers mixed with 3 mL of 4 × 10^−5^ M aqueous methyl orange solution. The samples were left in the dark for 24 h and then exposed to UV and visible light lamps. The decomposition of the methyl orange was checked by measuring the absorbance of its most intense peak (464 nm) every half-hour using a Jasco V-550 UV-VIS spectrometer (Jasco, Tokyo, Japan) for 240 minutes.

## 3. Results and Discussion

Figure 1 shows the TG/DTA curves for the thermal decomposition of the as-spun fibers in air and nitrogen. The decomposition process was continuous in both conditions. The DTA curves for the decomposition process in air showed exothermic peaks because of the decomposition reactions that occurred. In nitrogen, the fiber components decomposed without combustion, and the DTA curves showed endothermic peaks. The residual mass was higher when the fibers were decomposed in nitrogen due to the presence of unburnt carbon [15].

When the fibers were decomposed in air, there was a loss of mass of about 6% at around 100 °C due to the loss of water. The exothermic peaks between 300 and 350 °C (Figure 1a–e) are due to the decomposition and combustion of PVP, and AMT also begins to decompose at this stage. In Figure 1b–e, the exothermic peaks at around 400–480 °C are due to the degradation and decomposition of TiBALDH. In Figure 1a–d, the exothermic peak between 510 and 560 °C can be attributed to the burning of the carbon residue of PVP. The as-formed h-WO_3_ also transformed into m-MO_3_ in this stage [26]. The decomposition of the polymer and the degradation of the precursors finished at around 560 °C. The resulting fibers showed a significant increase in mass with an increase in the concentration of AMT. This is consistent with the theoretical knowledge, since AMT is about 94.1% WO_3_ by mass, while TiBALDH is 27.2% TiO_2_ by mass.

The SEM images of the as-spun fibers are shown in Figure 2. The diameter of the fibers increased with an increase in the AMT ratio. The diameter was 630–800 nm for 100% TiBALDH, 720–1050 nm for 90% TiBALDH, 920–1170 nm for 50% TiBALDH, 980–1200 nm for 10% TiBALDH and 1930–2950 nm for 0% TiBALDH.

As shown in Figure 3, the diameter of the fibers reduced after annealing. This can be attributed to the decomposition and degradation of PVP, TiBALDH, and AMT. The fibers had a diameter between 130 and 170 nm for 100% TiBALDH, 420 and 480 nm for 90% TiBALDH, 700 and 740 nm for 50% TiBALDH, 780 and 1020 nm for 10% TiBALDH, and 1610 and 1940 nm for 0% TiBALDH. The elemental composition and the specific surface area of the annealed fibers are shown in Table 1.

The FTIR spectra of the fibers are shown in Figure 4. For the as-spun nanofibers, the vibration peaks at around 3200 cm^−1^ are due to the stretching vibrations of –OH from water and –NH functional groups from AMT [27]. Peaks between 2980 and 2870 cm^−1^ are associated with the C–H stretching vibrations from the lactato ligands of TiBALDH. The peak at around 1700 cm^−1^ corresponds to the C=O stretching and the –NH bending vibrations in TiBALDH, while around 1430 cm^−1^ and 1470 cm^−1^ bands bending –CH bonds of CH_2_ in PVP and deformation mode of the NH_4_^+^ ion in AMT can be observed. C–O and C–N stretching bands of PVP can be seen in the range 1250–1220 cm^−1^. Stretching and bending vibrations of C=C can be seen at 1550 cm^−1^ and 980 cm^−1^, respectively [28]. Peaks at around 700–900 cm^−1^ can be attributed to W–O vibrations. The peak at around 550 cm^−1^ is assigned to Ti–O stretching motions [29]. After annealing C–H, C=O, C–N, and C=C, vibration peaks disappeared, demonstrating that the annealing process was successful. 

Figure 5 shows the XRD patterns of the as-spun and annealed fibers. The XRD pattern of the as-spun fibers did not show distinct diffraction peaks due to the amorphous nature of the as-spun fibers. The XRD pattern of 100% TiBALDH fibers was indexed to ICDD 04-022-6651. The pattern had peaks corresponding to the anatase form of TiO_2_. The diffraction peaks at 2θ = 25.3, 28.6, 48.1, 55.1 and 62.7 corresponded to (101), (112), (200), (105) and (204), respectively [30,31]. The XRD pattern of 0% TiBALDH fibers was indexed to ICDD 04-021-7427 and corresponded to the monoclinic form of WO_3_. The pattern showed the main characteristic peaks corresponding to (002), (020), (200) and (202) planes at 2θ = 23.1, 23.6, 24.4, and 34.2, respectively [32]. The XRD patterns for 90%, 50%, and 10% TiBALDH showed peaks corresponding to the anatase form of TiO_2_ and monoclinic forms of WO_3_.

Figure 6 shows the Raman spectra of the annealed fibers; 100% TiBALDH showed Raman active modes for anatase TiO_2_; 144 cm^−1^ (E_g_), 197 cm^−1^ (E_g_), 399 cm^−1^ (B_1g_), 515 cm^−1^ (A_1g_) and 630 cm^−1^ (E_g_) [19]. The peaks at 720 cm^−1^ and 805 cm^−1^ for 0% TiBALDH can be assigned to the O=W=O stretching vibrations characteristic of monoclinic WO_3_ [33]. The Raman spectra for annealed 10%, 50%, and 90% TiBALDH showed Raman peaks for both anatase TiO_2_ and monoclinic WO_3_.

The diffuse reflectance spectra in Figure 7 showed that the absorption edge of the fibers increased with an increase in the concentration of AMT. The increase in the absorption edge implies that the composite nanofibers can absorb light at a higher wavelength. Therefore, they can utilize light more efficiently during the process of photocatalysis [22]. The bandgap was calculated from the Tauc plots (Appendix A) using Kubelka-Munk theory. The bandgap decreased as the ratio of AMT increased, as shown in Table 2. This confirmed that the electrospinning process resulted in the coupling of TiO_2_ with WO_3_, hence a decrease in the bandgap. This is significant, as the decrease in the bandgap can enable the composite fibers to absorb light in the visible region of the electromagnetic spectrum.

Figure 8 illustrates the rate of photocatalytic degradation of methyl orange in the presence of annealed fibers in UV and visible light. In UV light, P25 showed higher photocatalytic activity than the annealed fibers. For the as-prepared fibers, 50% TiBALDH fibers showed the highest photocatalytic activity after 240 min, and its activity was very close to that of P25. This can result from the combination of TiO_2_ and WO_3_, which decreased the rate of recombination of photogenerated electrons and holes during the photocatalytic process. In visible light, 0% TiBALDH fibers had the greatest photocatalytic activity. For the composite fibers, the photocatalytic activity increased with the increase in the ratio of AMT. The fibers had increased absorption edges and could absorb light at a higher wavelength.

The reaction constants for the degradation of methyl orange was calculated using a pseudo first-order kinetics model. A plot of −ln(A/A^o^) against time gave a straight line (Appendix A), and the slope represented the apparent rate constant (k_app_) for the degradation reactions, as shown in Table 3 and Table 4. The k_app_ was consistent with the photocatalytic activity of the as-spun fibers. The fibers that showed better photocatalytic activity had higher K_app_ values. 

The photocatalytic efficiency of the 50% TiBALDH fibers compared favorably with other composite TiO2/WO3 fibers reported by other authors, as shown in Table 5.

## 4. Conclusions

TiO_2_/WO_3_ composite fibers were successfully synthesized by electrospinning, using water-soluble precursors and annealing. The fibers were characterized by TG/DTA, SEM-EDX, FTIR, XRD, and Raman. Annealed fibers had the anatase form of TiO_2_ and the monoclinic form of WO_3_. Different precursor ratios were used to investigate how the concentration of the precursors affected different properties of the fibers. The diameter of the fibers depended on the precursor ratio and increased with the increasing concentration of AMT. Composite fibers with 50% TiBALDH had higher photocatalytic activity in UV light than pure fibers. This can be as a result of the reduction in the rate of recombination of the charge carriers. The photocatalytic activity of the fibers in visible light increased with an increasing ratio of AMT. Coupling TiO_2_ with WO_3_ can increase the absorption edge of the fibers, hence making the fibers absorb light at a higher wavelength of the spectrum, thereby improving the photocatalytic property of TiO_2_.

## Figures and Tables

**Figure 1 nanomaterials-10-00882-f001:**
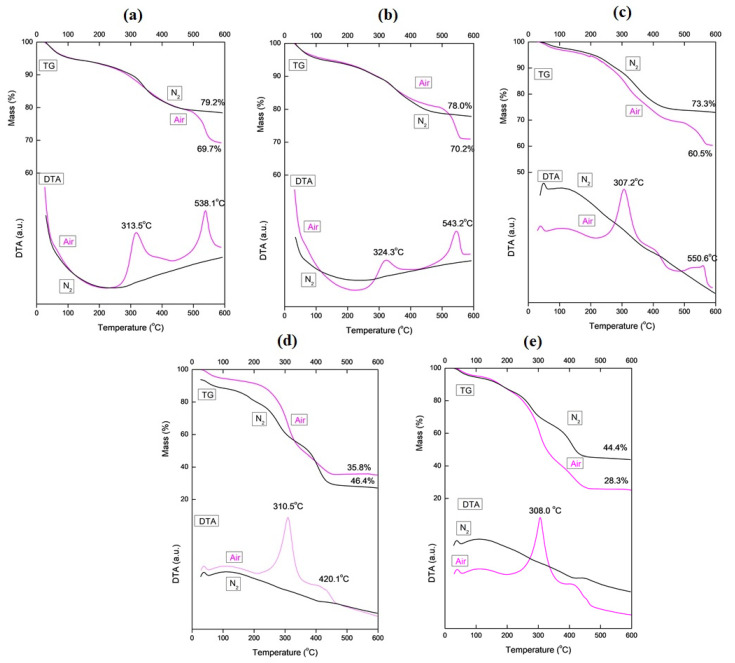
Thermogravimetry/differential thermal analysis (TG/DTA) curves of as-spun fibers annealed in air and nitrogen: (**a**) 0% titanium(IV) bis(ammonium lactate)dihydroxide (TiBALDH, (**b**) 10% TiBALDH, (**c**) 50% TiBALDH, (**d**) 90% TiBALDH and (**e**) 100% TiBALDH.

**Figure 2 nanomaterials-10-00882-f002:**
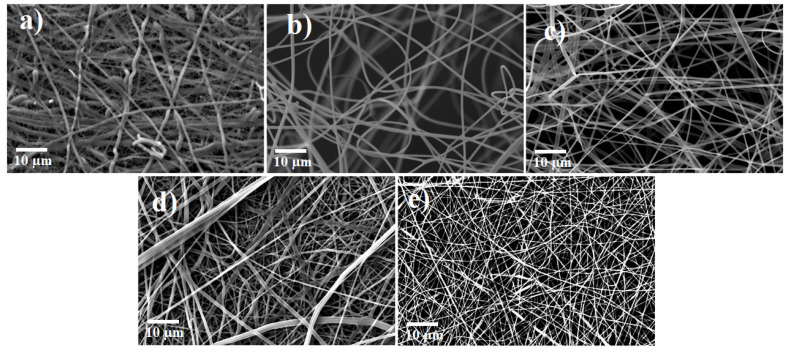
SEM images of as-spun fibers: (**a**) 0% TiBALDH, (**b**) 10% TiBALDH (**c**) 50% TiBALDH, (**d**) 90% TiBALDH, and (**e**) 100% TiBALDH.

**Figure 3 nanomaterials-10-00882-f003:**
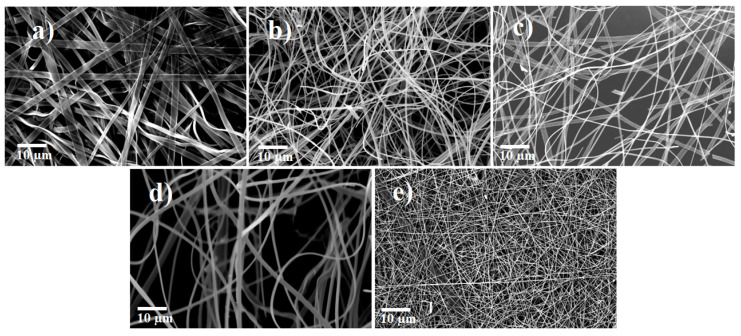
SEM images of annealed fibers (**a**) 0% TiBALDH, (**b**) 10% TiBALDH (**c**) 50% TiBALDH, (**d**) 90% TiBALDH, and (**e**) 100% TiBALDH.

**Figure 4 nanomaterials-10-00882-f004:**
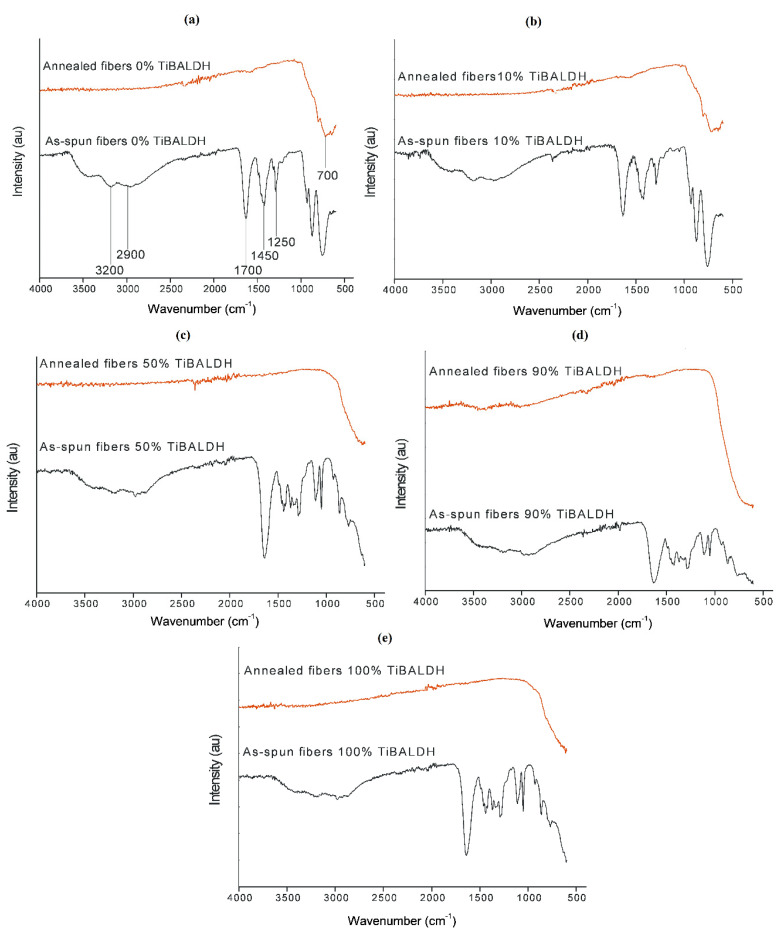
FTIR spectra of as-spun and annealed fibers: (**a**) 0% TiBALDH, (**b**) 10% TiBALDH, (**c**) 50% TiBALDH, (**d**) 90% TiBALDH and (**e**) 100% TiBALDH.

**Figure 5 nanomaterials-10-00882-f005:**
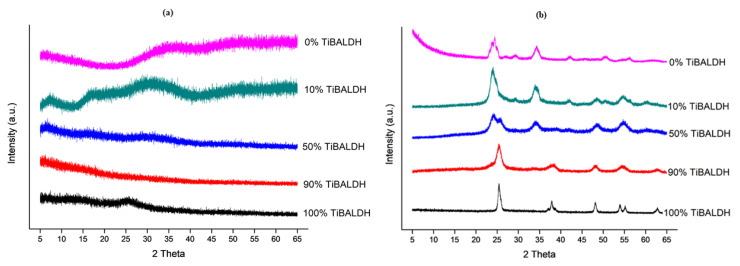
XRD patterns: (**a**) as spun fibers, and (**b**) as annealed fibers.

**Figure 6 nanomaterials-10-00882-f006:**
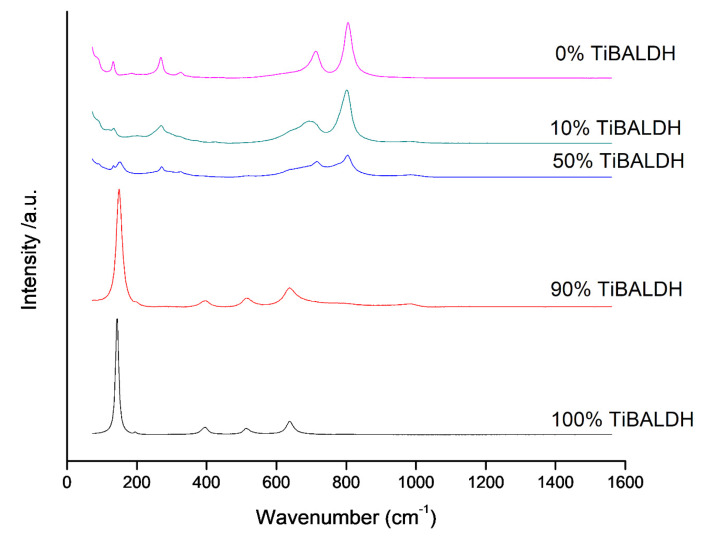
Raman spectra of annealed fibers.

**Figure 7 nanomaterials-10-00882-f007:**
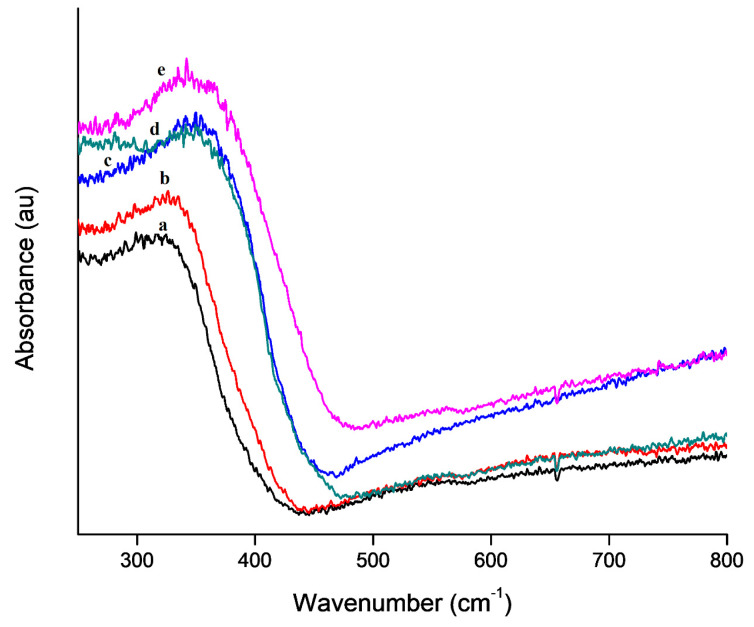
Diffuse reflectance UV-Vis spectra: (**a**) 100% TiBALDH, (**b**) 90% TiBALDH (**c**) 50% TiBALDH, (**d**) 10% TiBALDH, and (**e**) 0% TiBALDH.

**Figure 8 nanomaterials-10-00882-f008:**
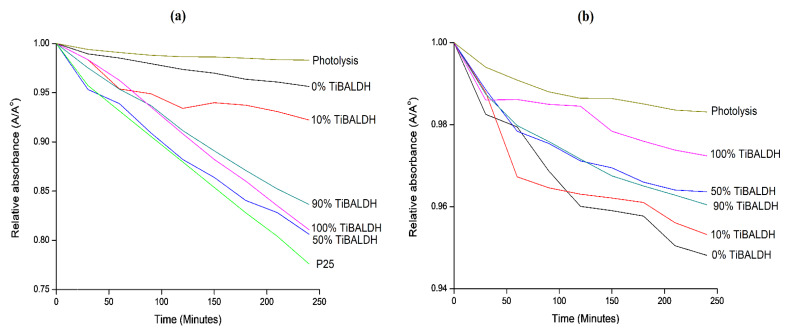
The degradation rate of methyl orange by annealed fibers in (**a**) UV light, and (**b**) visible light.

**Table 1 nanomaterials-10-00882-t001:** The diameter of as-spun and annealed fibers, elemental composition and specific surface area of annealed fibers.

Fibers	Diameter (d/nm)		Elemental Composition (wt %)			Surface Area (m^2^/g)
	as-Spun Fibers	Annealed Fibers	Ti	W	O	
100% TiBALDH	630–800	130–170	35.8		64.2	1.5
90% TiBALDH	720–1050	420–480	32.2	6.0	61.8	49.4
50% TiBALDH	920–1170	700–740	12.1	23.1	64.7	25.8
10% TiBALDH	980–1200	780–1020	7.3	28.8	63.9	11.3
0% TiBALDH	1930–2950	1610–1940		36.8	63.2	9.3

**Table 2 nanomaterials-10-00882-t002:** Bandgap values for the annealed fibers.

Fibers	100% TiBALDH	90% TiBALDH	50% TiBALDH	10% TiBALDH	0% TiBALDH
**Bandgap [eV]**	3.06	2.96	2.73	2.62	2.58

**Table 3 nanomaterials-10-00882-t003:** Reaction rate constant and r^2^ values for the photocatalytic degradation of methyl orange in UV light.

Sample	k_app_ (min^−1^)	r^2^
100% TiBALDH	0.0009	99.6
90% TiBALDH	0.0007	99.8
50% TiBALDH	0.0009	98.9
10% TiBALDH	0.0003	82.7
0% TiBALDH	0.0002	98.4
Bare Methyl Orange	0.00006	86.6
P25	0.001	99.8

**Table 4 nanomaterials-10-00882-t004:** Reaction rate constant and r^2^ values for the photocatalytic degradation of methyl orange in visible light.

Sample	k (min^−1^)	r^2^
100% TiBALDH	0.0001	86.1
90% TiBALDH	0.0002	91.4
50% TiBALDH	0.0001	87.2
10% TiBALDH	0.0002	78.1
0% TiBALDH	0.0002	92.0
Bare Methyl Orange	0.0006	86.0

**Table 5 nanomaterials-10-00882-t005:** Comparison of photocatalytic efficiency of various WO_3_/TiO_2_ nanofibers.

Photocatalyst	Method of Preparation	A/A°	Authors
WO_3_/TiO_2_ films	Electrospinning and deposition	0.85	Soares et al. [34]
WO_3_/TiO_2_ core shell nanofibers	Electrospinning and ALD	0.65	Szilagyi et al. [10]
WO_3_/TiO_2_-wood fibers	Hydrothermal synthesis	0.2	Gao et al. [35]
WO_3_/TiO_2_ nanoparticles	Hydrothermal sol-gel synthesis	0.2	Paula et al. [36]

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
