# Peer review of "Synthesis of TiO2/WO3 Composite Nanofibers by a Water-Based Electrospinning Process and Their Application in Photocatalysis"

_nanomaterials, 2020, doi:10.3390/nano10050882_

Round 1

Reviewer 1 Report

This is my review report for the manuscript titled "Synthesis of TiO2/WO3 Composite Nanofibers by 3Water-Based Electrospinning Process and Their Application in Photocatalysis"

The goal of the authors is to highlight the role of the electrospinning process in the forementioned composites making some variations in the amount of the precursors and proving its performance as photocatalyst. The whole manuscript presents the sufficient characterization: TG/TDA, XRD, SEM, EDX, FTIR and Raman. The document is well written, easy to follow and all the results are consistent and well explained.

However, the quality of the manuscript should be increased to be published in nanomaterials. My following comments will highlight some problems I met reading the manuscript and which must be tackle before submit a new version of the paper:

  • 2 Preparation and characterization of TiO2/WO3 fibers must be in bold.
  • Figures must be found in the same conditions, some are very small and other really big. The same happened with fonts. The authors should take care of this fact because it gives a sloppy appearance.
  • The scale bar of the SEM micrographs must be bigger, because they cannot be seen.

To end this review, could the authors provide XRD of the as spun fibers and discuss them? I find surprisingly how the annealing process can reduce the size of the fibers.

Reviewer 2 Report

The manuscript is describing the electrospinning of TiO2/WO3 composite nanofibers and their application in photocatalysis. The manuscript is interesting and can be accepted after addressing the following concerns.

  1. Figure 1 and 2 can be plotted together in a single graph for better comparison in air and nitrogen. Author can use different symbol/color for air and nitrogen.
  2. Figure 3 and 4 are of poor resolution. Need to modify with visible scale bars. Also, make all SEM images of fiber with varying compositions in identical magnification frame. 
  3. Table 1 shows some interesting results. 100% TiBALDH and 90% TiBALDH showed fiber diameter changed huge. Also, the BET data is surprising. Why does surface area reduce ~1 m2/g from ~50 m2/g when composition changes 10%? Any data to support there was significant reduction of porosity? Authors should explain. May be pore size distribution analysis from BET data help.
  4. Author should use identical color/symbol in two plots for TiBALDH composition. This will help to understand the relative comparison of two cases. Also, what was the degradation of MO in dark? Author should calculate and plot the rate of reaction for photocatalytic study. 
  5. A comparison table would be nice if authors show photocatalytic efficiency of MO degradation by their fibers as compared to other fibers.

Round 2

Reviewer 2 Report

The authors have addressed all the comments raised by the referees. This manuscript can now be accpeted.